# The Association between Executive Functions and Body Weight/BMI in Children and Adolescents with ADHD

**DOI:** 10.3390/brainsci11020178

**Published:** 2021-02-01

**Authors:** Ewa Racicka-Pawlukiewicz, Katarzyna Kuć, Maksymilian Bielecki, Tomasz Hanć, Anita Cybulska-Klosowicz, Anita Bryńska

**Affiliations:** 1Department of Child and Adolescent Psychiatry, Medical University of Warsaw, Żwirki i Wigury 61, 02-091 Warsaw, Poland; abrynska@interia.pl; 2Department of Psychology, SWPS University of Social Sciences and Humanities, 03-815 Warsaw, Poland; kkuc@swps.edu.pl (K.K.); mbielecki@swps.edu.pl (M.B.); 3Institute of Human Biology and Evolution, Faculty of Biology, Adam Mickiewicz University, 61-614 Poznan, Poland; tomekh@amu.edu.pl; 4Laboratory of Neuroplasticity, Nencki Institute of Experimental Biology of Polish Academy of Sciences, 02-093 Warsaw, Poland; a.cybulska@nencki.edu.pl

**Keywords:** ADHD, executive functions, obesity

## Abstract

Despite the increasing body of research on Attention Deficit Hyperactivity Disorder (ADHD), the results of the studies assessing the relationship between executive function deficit and the risk of obesity in people with ADHD are incongruent. Our study aimed to assess the relationship between measures of executive functions and body weight and Body Mass Index (BMI) in children and adolescents with ADHD and control subjects. The study group consisted of 58 subjects aged from 8 to 17 years with ADHD. The Control group consisted of 62 healthy age and sex-matched participants from primary and secondary schools. Weight, height, and BMI measurements were standardized. The Sustained Attention to Response Test (SART) and the Attention Network Test (ANT) were used to assess executive functions. Based on the analysis of the correlation and analysis of moderation, we found that subjects with higher weight in the study group presented a lower efficiency of the inhibition processes and gave more impulsive and incorrect answers. The occurrence of impulsive reactions might contribute to the risk of excessive weight in children and adolescents with ADHD.

## 1. Introduction

In recent years, hypotheses regarding the association between attention deficit hyperactivity disorder symptoms and overweight/obesity have been verified. Hypotheses concerned, among others: genetic factors [1,2,3,4], eating habits [5,6], the influence of comorbid disorders [7,8,9,10,11], sleep disorder [12], or physical activity and lifestyle, i.e. the number of hours spent in front of the TV [13,14,15,16,17,18,19]. Some studies have assessed the influence of neurocognitive deficits on the occurrence of overweight/obesity in patients with Attention Deficit Hyperactivity Disorder (ADHD) [3,20,21,22].

Despite the increasing body of research on ADHD, the results of the studies assessing the relationship between executive function deficit and the risk of obesity in people with ADHD are incongruent. In the study of Graziano et al. [20], carried out in a group of 80 children and adolescents with ADHD between 4.5 and 18 years, subjects who achieved lower results in neuropsychological tests were characterized by higher z-Body Mass Index (z-BMI) values. Hanć et al. [3] did not show significant differences in neuropsychological tests in ADHD patients with or without overweight (a group of 109 ADHD boys aged from 7 to 17 years). Similarly, in Choudhry’s et al. study [2] carried out on a group of 284 children and adolescents with ADHD between 6 and 12 years and Van der Oords’s et al. study [21] in the group of 39 adolescents and adults with obesity aged from 17–68 years, such relationship was not confirmed. It should be emphasized, however, that only in the Van der Oord study obtained results were referred to the control group.

This study aimed to assess the relationship between measures of executive functions and body weight, and BMI in children and adolescents with ADHD and control subjects. 

We hypothesized that there is an association between the intensity of ADHD symptoms, such as inattention, impulsivity, and executive function deficits, and excessive weight, which would be of a broader range in the ADHD subgroup compared to healthy control.

Several essential features distinguish our work from the published studies. First, we used computerized behavioral procedures known to be valid measures of ADHD-related executive deficits in the Sustained Attention to Response Task (SART) (Robertson et al. [22]) and the Attention Network Task (ANT) (Fan et al. [23]) tasks instead of the standardized paper and pencil or performance tests or tasks tapping other attentional processes (i.e., Continous Performance Tasks (CPT)). Second, in the regression analyses, we controlled for the effects of age, which is of key importance considering the broad age range in most tested samples. Finally, our regression models allowed the estimation of the moderating effects of the diagnosis on the relationship between executive functioning and body weight indices.

## 2. Materials and Methods

### 2.1. Study Group

The clinical group consisted of 60 patients from the outpatient psychiatry clinic, aged from 8 to17 years (M = 13.17 years, SD = 1.98, min = 8.36, max = 16.51), from which 58 were qualified for further statistical analysis (two were excluded because of extremely low accuracy in cognitive tasks). The diagnosis was performed according to the diagnostic criteria of the Diagnostic and Statistical Manual of Mental Disorders (4th ed., text rev.; DSM-IV-TR; American Psychiatric Association, 2000 [24]) for one of the three ADHD types: predominantly inattentive or impulsive/hyperactive or combined type as previously described [10]. For the comorbid diagnosis, supplements of K-SADS-PL (Kiddie-Schedule for Affective Disorders and Schizophrenia for School-Age Children-Present and Lifetime version; K-SADS-PL ver.1.0; Kaufman et al., [25]) were administered by a child psychiatrist, and the diagnosis was based on the International Statistical Classification of Diseases and Related Health Problems, 10th Revision (ICD-10) diagnostic criteria (World Health Organization, 1994) [26]. The diagnosis was performed during at least three appointments, during which children also underwent physical, neurological, and developmental examination. Exclusion criteria included comorbid disorders which could cause weight gain (e.g., diabetes, dysfunction of the thyroid gland, as well as psychiatric disorders, e.g., depressive disorder) and any problems that could result in limited collaboration or affect the sensory aspects of task performance (e.g., defect vision, hearing loss).

Control group consisted of 62 healthy age and sex-matched children and adolescents from primary and secondary schools, aged from 8 to 17 years (M = 13.70 years, SD = 2.0, min = 9.44, max = 17.04). The process of control group recruitment was based on a questionnaire filled by the participants’ parents, which provided information about the child’s physical and mental condition. Based on the obtained information, we included participants who did not exhibit any attentional problems, neurological or psychiatric diagnosis, dyslexia, previous brain injury with the loss of consciousness, and had no close family members (parents, siblings) with ADHD/ADD diagnosis or any disease that could cause weight gain (same criteria as for the ADHD group).

All participants received 150 Polish Zloty (PLN) for the study.

### 2.2. Anthropometric Measurements 

Bodyweight was measured with a medical scale (Radwag PUE c/31) with an accuracy of ±100 g. Height was measured using a Harpenden anthropometric instrument with an accuracy of 0.1 cm according to the standard technique [27,28] by trained medical staff. The examined children were weighed in their underwear. The measurements were performed between 9 a.m. and 4 p.m. BMI was calculated on the basis of body weight and height. Height, weight, and BMI were later analyzed as norm-referenced standardized scores according to sex and age using the most recent available growth references for the Polish population (published by Kułaga et al. (2011) [29]). 

### 2.3. Calendar Age

The calendar age at the time of the test was calculated on the basis of the date of examination and the date of birth of the patient. The age was determined in annual intervals.

### 2.4. Cognitive Tests

Two cognitive tests were used: the Sustained Attention to Response Test (SART; Robertson et al. [22]) and the Attention Network Test (ANT; Fan et al. [23]). All ADHD patients did not take the stimulant medication 24 h before testing. Behavioral data were obtained during one measurement session carried out in the laboratory of the SWPS University of Social Sciences and Humanities in Warsaw, Poland. The total measurement time (including psychophysiological measurements) included approximately 2.5 h.

#### 2.4.1. Sustained Attention to Response Test

SART (Robertson et al. [22]) allows assessing sustained attention and response inhibition (propensity to give impulsive responses). Subjects were presented with a sequence of randomly selected numbers (from 1 to 9) on the computer screen. Each digit was presented for 250 ms and followed with a mask presented for 900 ms. The respondent had to press the button on the keyboard in response to all numbers appearing on the screen (‘go’ trial) except the ‘3’ number (‘no go’ trial). The test included 255 attempts, out of which 230 were ‘go’ trials and 25 ‘no go’ trials. The main part of the study took place after a short training consisting of 12 trials. Collected measures included accuracy and reaction time (RT) indices, such as commission and omission errors, mean RT for correct go-trials responses, and a coefficient of variation (index of reaction variability computed as the standard deviation of reaction times in go-trials divided by their mean). Due to extremely low accuracy, one additional subject from the control group and one additional subject from the ADHD group were dropped from statistical analysis.

#### 2.4.2. Attention Network Test

ANT (Fan et al. [23]; Posner [30]) was used to assess the effectiveness of three attentional networks, as defined by Posner [30], responsible for the following aspects of processing: alerting, orienting, and executive control. This task consisted of a series of trials in which a central target arrow (signal) appeared on the screen surrounded by other arrows (so-called flankers). Flankers may be pointing in the same direction as the central arrow (signal) or in the opposite direction. Participants were instructed to indicate (using computer-mouse buttons) whether the central arrow (signal) on the screen was pointing to the left or right. In incongruent trials, participants had to refrain from the imposing but incorrect reaction suggested by the flankers.

Additionally, the target could be preceded by the presentation of one of the three cues: central (a star appearing in the middle of the screen), double (two stars), or peripheral (informing about the location of the signal). The cues provided participants with information about the temporal and spatial aspects of the upcoming target. The reaction time and accuracy of the reaction were recorded. The effectivity of attentional networks was determined according to standard formulas (Fan et al. [23]). 

### 2.5. Statistical Methods

Intergroup comparisons for continuous variables were performed using the Student’s *t*-test with Welch’s correction or the Mann–Whitney test depending on the distribution of analyzed variables. The Pearson correlation coefficient was used to assess the relationship between standardized values of height, weight, and BMI with age. The effect of the cumulated dose of drugs on the risk of overweight and obesity in the study group was assessed using logistic regression analysis. 

Correlation analysis for cognitive performance indicators was conducted using the Spearman Rank Correlation (Spearman rho) correlation coefficient due to the lack of normal distribution of many variables. Moderation analysis was performed using linear regression. The significance of the effects was determined based on heteroscedasticity-consistent standard errors (HC3). 

Additionally, following the postulated link between the decreased executive and inhibitory control and obesity in ADHD, we ran a series of moderation analyses. Key performance indices were used as predictors, and their relationship with participants’ weight was moderated by Group (ADHD vs. control). The quality of executive/inhibitory control was captured using three indices considered most valid in the context of ADHD-specific deficits, that is, executive control scores from ANT, and, Coefficient of Variation, and Accuracy in no-go trials from SART. 

As the cognitive performance scores were not age-referenced, in each of the analyses, they were regressed on age before entering the model. Two strongly skewed indices in SART were normalized using square-root transformation. Interpretation of lower-order effects in regression was facilitated by sum-contrast coding of the grouping variable (ADHD = −1, control group = 1). Moreover, in the modelling of norm-referenced body mass, centred height was added to the model as a necessary predictor of body weight not taken into account in age and sex referenced norms.

Results were considered statistically significant at a significance level of *p* < 0.05. R environment (version 3.5.3, R Core Team [31]) was used in all statistical analyses. 

## 3. Results

Boys constituted 79.3% of the ADHD study group (46 out of 58 patients). Forty-one (70.7%) patients (35 boys, 6 girls) had combined subtype of ADHD, *n* = 16 (27.6%) (10 boys, 6 girls) predominantly had the inattentive subtype, and *n* = 1 (1.7%) boy had predominantly the hyperactive-impulsive subtype. In 43 (74.1%) patients at least one comorbid condition was diagnosed. The most common were oppositional defiant disorder (ODD) (*n* = 30, 51.7%). Specific developmental disorders of scholastic skills were found in *n* = 21 patients (36.2%), tic disorder in *n* = 2 (3.4%) patients, and conduct disorders in *n* = 1 patient (1.7%). 

Most of the respondents at the time of study received medication (*n* = 53, 91.4%, including 42 boys). The most frequently used were methylphenidate (MPH) osmotic release oral system (OROS) (74.1% of the sample) and sustained-release (SR) (17.2% of the sample). The average daily dose of methylphenidate OROS was 36.8 mg/day (min 18 mg/day, max 54 mg/day), methylphenidate SR 27.0 mg/day (min 10 mg/day, max 50 mg/day). Adjusted for a holiday break, the average duration of pharmacological treatment for OROS methylphenidate was 1096.9 days (min 73 days, max 2777 days), and for methylphenidate SR, the average was 1188 days (min 583 days, max 2139 days) (Table 1).

### 3.1. The Values of Height, Body Weight, and BMI

Norm-referenced (Kułaga et al., 2011) [29] standardized mean values of height–(zH-score for height), weight (zW-z-scores for weight), and BMI (zBMI-z-scores for Body Mass Index) are shown in Table 2. No differences in the standardized mean values of zH, zW, zBMI between the ADHD group and the control group were found (for the whole groups and the subgroups of boys).

### 3.2. Results of the Cognitive Tasks

Significant differences between the study and the control group were obtained in SART in terms of accuracy in go trials, accuracy in no-go trials (requiring response inhibition), and the coefficient of variation (Table 3).

Significant differences between the study and the control group in ANT were observed in average reaction times, accuracy in both congruent and incongruent trials, and the index of Executive control efficiency (Table 4).

### 3.3. The Association between Mean Norm-Referenced Standardized Height, Body Weight, and BMI and the Results of Cognitive Tasks

The results of accuracy in ‘no go’ trials of SART, requiring response inhibition to stimuli in the study group, were negatively correlated with the mean standardized bodyweight (*p* = 0.042, rho = −0.27) (Table 5).

### 3.4. ADHD Diagnosis and Executive Functioning as Predictors of Body Weight

The regression analyses did not reveal significant effects of moderation for the executive score in ANT, both for body weight and BMI. In terms of SART, statistically significant results were obtained for: (1) Coefficient of Variation and (2) Accuracy in no-go trials. Both moderation effects were obtained in analyses where norm-referenced standardized body mass was the dependent variable. Table 6 and Table 7 present the results of the regression analysis for both indices. Figure 1 and Figure 2 present the key interaction effects for each of the models.

In the case of the model described in Table 7 and illustrated in Figure 1, the interaction effect indicates a different relationship between the quality of SART task performance measured by the coefficient of variation and body weight. In the control group, the worse performance of the task was accompanied by lower body weight, whereas in the ADHD group, this relationship was the opposite direction.

A qualitatively similar effect was obtained in the second analysis, where the more effective inhibition processes represented by a higher accuracy in the no-go trials were accompanied by higher body weight in the control group and lower in the ADHD group.

## 4. Discussion

### 4.1. The Results of Cognitive Tests

In our study, participants with ADHD showed less effective processes of sustained attention, less effective response inhibition, and lower stability of attention compared to the control group. In SART:(1)Participants with ADHD made significantly more errors in both ‘go’ trials, which showed their less effective sustained attention, as well as in ‘no-go’ trials, which demonstrated less effective response inhibition and greater impulsivity. These results are consistent with the observations of Johnson et al. [32], O’Connell et al. [33], Shallice et al. [34], and Wodka [35], which in turn is in line with the hypothesis that deficit in response inhibition is a primary mechanism causing ADHD symptoms (Barkley [36]);(2)Participants with ADHD had significantly higher values of the Coefficient of Variation, which means lower stability of reaction times, thus less stability of attention processes (Coefficient of Variation is a measure of intra-subject variability, which is characteristic of ADHD (Castellanos and Tannock [37]). These results are consistent with the results of other studies (Bellgrove et al. [38]; Castellanos et al. [39]; Epstein et al. [40]; Gómez-Guerrero et al. [41]; Hervey et al. [42]; Hynd et al. [43]; Klein et al. [44]; Leth-Steensen et al. [45]). It is believed that the Coefficient of Variation better characterizes people with ADHD than the accuracy of performing tasks or the mean reaction time (Klein et al. [44]). Furthermore, intra-subject variability has been shown to be hereditary, both for people with ADHD and people from the control group (Andreou et al. [46]; Kuntsi et al. [47]; Kuntsi and Stevenson [48]);

They responded faster to stimuli and had a lower mean reaction time in response to ‘go’ trials-this difference was not statistically significant. In addition, in our study, participants with ADHD compared to the control group showed pronounced deficits in the executive control, i.e., they were more susceptible to distraction, were characterized by less stable attention processes, and less effective inhibition of the attention orienting. In the ANT:(1)Participants with ADHD had significantly longer time in response to incongruent stimulus presentation, which indicated less effective inhibition of the attention orienting and greater distractibility. This is consistent with the results of studies of Konrad et al. [49], Johnson et al. [50], and Mullane et al. [51].(2)Participants with ADHD had significantly longer total reaction time in the task, which meant less effective attention processes during the whole task. This result is consistent with the results of studies of Johnson et al. [50] and Kratz et al. [52]. On the other hand, in the study of Adolfsdottir et al. [53], there were no significant differences in reaction time in the group of ADHD patients and the control group.(3)Participants with ADHD achieved a lower accuracy in congruent trials, which indicated worse stability of attention processes and lower accuracy in incongruent trials, which indicated greater sensitivity to distractors.(4)Participants with ADHD did not differ from the control group in terms of Alerting and Orienting, which indicated a similar level of attention span and similar time to redirect attention to the stimulus (similar to the study of Adolfsdottir et al. [53]). These results are in line with the assumptions in which executive control is a superior function related to prefrontal and frontal areas (dopaminergic and noradrenergic pathways), which explains the lack of evident deficits in alertness and orienting of attention ADHD (Fan et al. [23]). Different results were obtained by Mullane et al. [51] and Johnson et al. [50], who showed the presence of significant differences in the alertness of attention in children with ADHD compared with control groups.

### 4.2. The Association between Mean Standardized Height, Body Weight, and BMI and the Results of Cognitive Tasks

Correlation analysis and moderation analysis were performed to assess the associations. The results obtained in the ADHD group for accuracy in ‘no-go’ trials in SART, requiring response inhibition to stimuli, correlated negatively with standardized body weight, in relation to the standards of Kułaga et al. This indicated that children and adolescents with a higher body weight presented lower efficiency of inhibition processes and provided more incorrect and impulsive responses. A similar conclusion can be formulated based on the results of the moderation analysis-a significant effect was obtained for (1) accuracy in ’no-go’ trials and (2) Coefficient of Variation. Both moderation effects were obtained for analyses where standardized body mass was the dependent variable, which meant that in the study group, better performance in the ‘no go’ trials and greater stability of attention processes were associated with lower standardized body weight.

The results of the moderation showed an interesting discrepancy in the models of the relationship between the effectiveness of the inhibition process and weight. The relationship in the control group was significantly different than in the ADHD group. It may suggest that excessive weight in healthy children and adolescents is conditioned by other variables than in children and adolescents with ADHD. There were no statistically significant correlations for the executive score obtained in the ANT and body weight and BMI both in the study group and in the control group. This lack of significant relationships with ANT indices might, to some extent, reflect moderate reliability of network efficiency scores (Macleod et al. [54]), which, in turn, affects the statistical power of analyses.

Rapid-response impulsivity, also referred to as response inhibition or impulsive action, is one of the dimensions of impulsivity [55], which characterizes ADHD and is investigated using go/no go task as in SART. In our study, subjects with higher weight in the ADHD group presented a lower efficiency of the inhibition processes and gave more impulsive and incorrect answers in SART, which may suggest that the problem of excessive weight might be associated with impulsivity in this group. In contrast, no similar relationship was detected for the executive score measured by ANT.

## 5. Conclusions

The obtained results may suggest that the problem of excessive body weight in children and adolescents with ADHD is more clearly associated with the severity of impulsivity symptoms rather than attention deficits. 

However, they do not allow for an unambiguous determination whether the body weight increases as a consequence of less effective inhibition, and thus, for example, impulsive food intake, or whether inhibition becomes less effective with increasing body weight, or maybe we are dealing with both phenomena. What is the role of attention deficits?

## Figures and Tables

**Figure 1 brainsci-11-00178-f001:**
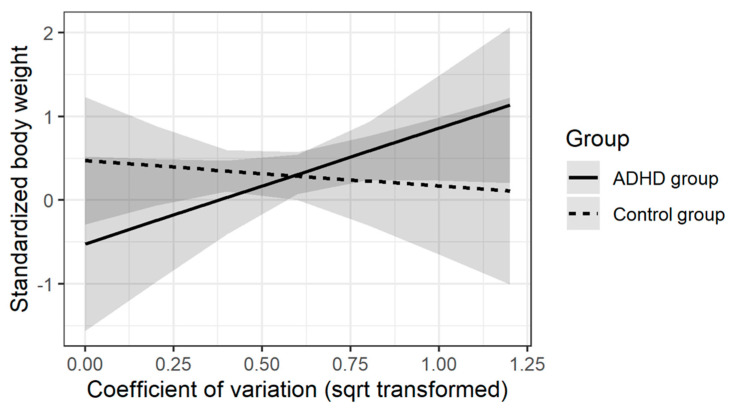
Moderating effects of Coefficient of Variation in Sustained Attention to Response Test (SART) task on the relationship between the ADHD diagnosis and norm-referenced standardized body weight. Note: Coefficient of variation (sqrt transformed)—Square root transformed Coefficient of variation.

**Figure 2 brainsci-11-00178-f002:**
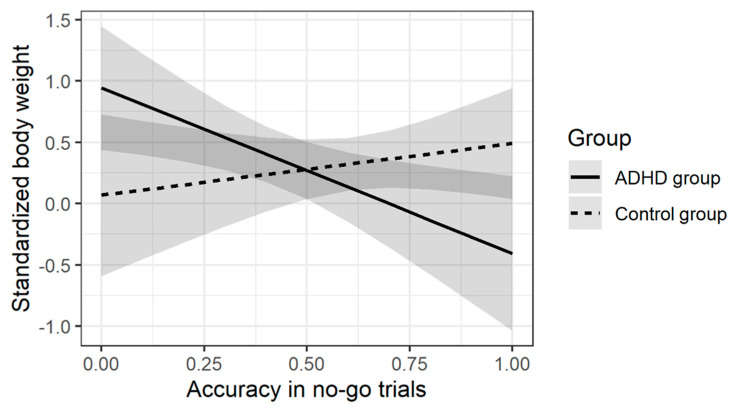
Moderating effects of accuracy in no-go trials in SART task on the relationship between the ADHD diagnosis and norm-referenced standardized body weight.

**Table 1 brainsci-11-00178-t001:** Dosage of metylphenidate in ADHD patients and adjusted treatment duration.

	*n*	M	Min	Max	SD
Adjusted treatment duration (days)	43	1096.9	73	2777	584
Methylphenidate OROS dose (mg)	43	36.8	18	54	12.4
Adjusted treatment duration (days)	10	1188	583	2139	575.6
Methylphenidate SR dose (mg)	10	27	10	50	12.5

**Table 2 brainsci-11-00178-t002:** Norm-referenced standardized mean values of height (zBV), weight (zBW), and Body Mass Index (BMI) (zBMI) based on Kułaga et al. (2011) growth charts and row BMI scores–comparison in ADHD and control groups [29].

	Control M	ADHD M	*t*	*df*	*p*	Control *n*	ADHD *n*	Control SD	ADHD SD
zBV (♂ + ♀)	0.43	0.22	1.16	118	0.25	62	58	0.97	1.03
zBW (♂ + ♀)	0.43	0.34	0.53	118	0.60	62	58	0.86	1.07
zBMI (♂ + ♀)	0.33	0.34	−0.04	118	0.97	62	58	0.84	1.18
BMI (♂ + ♀)	20.8	20.5	0.39	118	0.70	62	58	2.92	4.21

**Table 3 brainsci-11-00178-t003:** Comparison of Control and ADHD groups in Sustained Attention to Response Test (SART).

	Control (*n* = 61)				ADHD (*n* = 57)				
	Min	Med	Max	IQR	Min	Med	Max	IQR	*p*
Mean Reaction time in go trials	203	375	618	147	157	376	571	139	0.477
SD of RTs in go trials	46	131	359	61	77	182	416	100	<0.001
Coefficient of variation	0.10	0.33	0.93	0.15	0.23	0.49	1.17	0.24	<0.001
Accuracy in go trials	0.75	0.99	1.00	0.03	0.63	0.93	1.00	0.10	<0.001
Accuracy in no go trials	0.08	0.64	1.00	0.32	0.00	0.44	0.92	0.32	<0.001

Note: IQR—interquartile range.

**Table 4 brainsci-11-00178-t004:** Comparison of Control and ADHD groups in Attention Network Test (ANT).

	Control (*n* = 62)				ADHD (*n* = 57)				
	Min	Med	Max	IQR	Min	Med	Max	IQR	*p*
Total Reaction time	444	596	891	150	485	662	889	134	0.001
Accuracy in congruent trials	0.96	1.00	1.00	0.01	0.81	0.99	1.00	0.02	0.004
Accuracy in incongruent trials	0.73	0.96	1.00	0.04	0.24	0.94	1.00	0.06	0.006
Alerting	−52.75	25.84	129.40	37.51	−66.38	38.68	142.00	43.71	0.092
Orienting	−26.31	58.17	115.90	46.33	−24.12	55.18	127.80	38.76	0.411
Executive	29.81	98.55	242.50	43.74	−32.61	130.0	307.20	56.20	<0.001

Note: IQR—interquartile range.

**Table 5 brainsci-11-00178-t005:** Correlations between the performance indices in ANT and SART tasks and norm-referenced standardized values of body weight (zBW) and BMI (zBMI) in each of the groups.

		ADHD Group					Control Group				
			zBW		zBMI			zBW		zBMI	
	Task Performance Indices	*n*	rho	*p*	Rho	*p*	*n*	Rho	*p*	Rho	*p*
ANT	Total Reaction time	57	0.00	0.976	0.04	0.756	62	−0.02	0.896	0.07	0.579
	Accuracy in congruent trials		0.16	0.248	0.24	0.073		0.14	0.295	0.21	0.107
	Accuracy in incongruent trials		0.05	0.710	0.11	0.433		0.00	0.990	−0.05	0.681
	Alerting		0.10	0.440	0.05	0.713		−0.13	0.328	−0.06	0.657
	Orienting		0.05	0.722	0.10	0.477		0.05	0.723	0.06	0.655
	Executive		0.07	0.583	0.12	0.389		−0.04	0.778	0.10	0.427
SART	Mean Reaction time in go trials	57	−0.06	0.634	−0.03	0.837	61	0.11	0.380	0.08	0.542
	Coefficient of variation		0.17	0.196	0.09	0.483		−0.24	0.060	−0.13	0.334
	Accuracy in go trials		−0.20	0.137	−0.12	0.383		0.22	0.088	0.01	0.933
	Accuracy in no−go trials		−0.27	0.042	−0.21	0.119		0.24	0.063	0.20	0.129

Note: Norm-referenced standardized values of body weight (zBW) and Body Mass Index (zBMI) based on growth charts by Kułaga et al. (2011) [29].

**Table 6 brainsci-11-00178-t006:** Regression coefficients for a model predicting norm-referenced standardized body weight, including interaction effects of Coefficient of Variation in SART task and ADHD diagnosis.

	Beta	SE	*t*	*p*
Intercept	0.33	0.32		
Transformed CoV	−0.55	0.23	−0.24	0.81
Group (ADHD = −1, Control = 1)	0.65	0.33	1.98	0.050
Height (centered)	0.02	0.01	3.85	<0.001
Transformed CoV × Group	−0.51	0.24	−2.17	0.032
Model statistics	F(4, 113) = 7.12, *p* < 0.001, *R*^2^ = 0.15 Adjusted *R*^2^ = 0.12, interactionΔ *R*^2^ = 0.03			

Note: Transformed CoV—Square root transformed Coefficient of Variation computed for SART task.

**Table 7 brainsci-11-00178-t007:** Regression coefficients for a model predicting norm-referenced standardized body weight, including interaction effects of accuracy in no-go trials in SART and ADHD diagnosis.

	Beta	SE	*t*	*p*
Intercept	0.26	0.08		
Accuracy in no-go trials	−0.04	0.09	−0.43	0.66
Group	0.01	0.08	0.06	0.950
Height (centered)	0.02	0.01	3.57	0.001
Accuracy in no-go trials× Group	0.22	0.09	2.42	0.017
Model statistics	F(4, 113) = 6.54, *p* < 0.001, *R*^2^ = 0.17 Adjusted *R*^2^ = 0.14, interaction Δ*R*^2^ = 0.05			

## Data Availability

The data presented in this study are available on request from the corresponding author.

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
