# Peer review of "The Association between Executive Functions and Body Weight/BMI in Children and Adolescents with ADHD"

_brainsci, 2021, doi:10.3390/brainsci11020178_

Round 1

Reviewer 1 Report

In this manuscript, the authors investigated the association between several executive functions as evaluated by SART and ANT and body weight and BMI in children and adolescents with ADHD. The association between obesity and executive function deficits remains unclear in the literature and the current study may fill this gap. In line with previous reports, the authors confirmed compromised executive functions in subjects with ADHD compared to health controls. However, their main results on the association between executive functions and obesity are not well presented and explained; specifically, I have several major concerns with this manuscript.

1), BMI (or its categories, normal vs over weight) rather than body weight should be the primary index. Here it seems the authors failed to identify any significant association regarding BMI and instead focused their analysis on body weight. This should be sufficiently justified.

2), For their main analysis, the authors standardized BMI measures, normalized reaction time indices, and regressed some cognitive scores on age. It is understandable this is for the purpose of improving interpretability of the results, however, since linear and logistic regression do not make assumptions about the Gaussian distribution of data, for this study perhaps standardization rather than normalization and other treatments is preferred. Furthermore, line 102-104, height, weight, and BMI measures are not standardized in this sample alone but in reference to the national population? Given the relatively small sample size of this study, I am curious whether this standardization works.

3), The authors should specify the criteria of obesity in the methods. What is the difference between the criteria of Kulaga and Palczewska?

4), For the linear regression analysis, it seems the authors handily picked up several independent variables without justification. Based on the correlation results, one would expect that the authors include accuracy in go trials but this was not the case.

5), The contribution of the interaction term in the regression is unclear, the authors should report the R square change due to the interaction term.

Author Response

Dear editors and reviewers,

We would like to thank you for all the insightful comments on our manuscript “The association between executive functions and body weight/BMI in children and adolescents with ADHD”. All of these comments were very helpful and guided our efforts to substantially revise and improve our paper. We did our best to make relevant corrections. The detailed responses are provided below each of the reviewers’ remarks.

1. BMI (or its categories, normal vs over weight) rather than body weight should be the primary index. Here it seems the authors failed to identify any significant association regarding BMI and instead focused their analysis on body weight. This should be sufficiently justified.

For the study purpose, the analyses of both norm-referenced weight and BMI were important. As our sample represented a wide age range and both sexes, it was crucial to include norm-referenced scores allowing us to explain the effects of height, age and sex. We were interested in describing the complex interplay between ADHD diagnosis, executive functioning, and body weight which would not become visible without a proper control of other factors. At the same time, while understanding the clinical importance and consequences of overweight and obesity diagnosis, these were not our primary variables of interest because of two main reasons. Firstly, considering the observed prevalence and the group size, focusing on the dichotomous outcomes would not offer us sufficient power in detecting the interaction effects. Also, in our opinion, a methodological approach focused on continuous outcomes may be more suitable for identifying excessive weight in the pediatric population in which obesity may not be fully expressed yet. We did our best to clarify the methodological choices by significantly restructuring the paper.

2. For their main analysis, the authors standardized BMI measures, normalized reaction time indices, and regressed some cognitive scores on age. It is understandable this is for the purpose of improving interpretability of the results, however, since linear and logistic regression do not make assumptions about the Gaussian distribution of data, for this study perhaps standardization rather than normalization and other treatments is preferred. Furthermore, line 102-104, height, weight, and BMI measures are not standardized in this sample alone but in reference to the national population? Given the relatively small sample size of this study, I am curious whether this standardization works.

In the whole paper, we clarified the description of the norm-referenced weight scores. Weight results are expressed using Z-scores, which were not obtained by a simple within-sample statistical transformation, but are based on growth charts (as described in the updated method section). We hope that the term “norm-referenced standardized weight” will be more adequate in this context. To address other remarks, we significantly modified the paragraphs describing the moderation analyses.

We fully agree that the normality of predictors distribution per se is not a necessary condition of valid regression. Still, extremely strongly skewed RT indices in SART had to be square-root transformed before including into the models.

3. The authors should specify the criteria of obesity in the methods. What is the difference between the criteria of Kulaga and Palczewska?

Our manuscript is a part of a bigger project in which we focused, inter alia, on occurrence of overweight and obesity in ADHD children and adolescents according to different diagnostic criteria. However, it wasn’t our goal to report this data in this manuscript. Following your remarks and after carefully consideration of the (minor) differences observed between the correlational data, we decided to focus our paper only on the more recent growth charts by KuÅ‚aga et al. (based on data collected from 2007 to 2009). Charts published by Palczewska et al. use measurements collected earlier, between 1996 and 1999, however at the time of study they were still in use in some clinical settings. We also erased the obsolete sentence about the diagnosis of obesity according IOTF criteria from the abstract.

4. For the linear regression analysis, it seems the authors handily picked up several independent variables without justification. Based on the correlation results, one would expect that the authors include accuracy in go trials but this was not the case.

Following the reviewer’s suggestion, we have re-written the paragraph describing the choice of predictors. We would like to underline that our decision to test these particular measures (executive network efficiency from ANT and two inhibition measures from SART) was both theoretically driven and supported by the literature on differences between healthy and ADHD populations. Thus, the choice of indices was not results-driven. The full set of correlational results is available for descriptive purposes and transparency.

Referring to the accuracy in go trials, it is important to mention that the focal research questions of the paper are related to executive functions. The accuracy in go trials of SART is not a performance index having a clear theoretical meaning. Participants not reacting in all the go trials (and having lower accuracy) should also have a weaker propensity to react in no-go trials and fewer problems with inhibition. Considering the lack of relevant literature, expected ceiling effects (actually observed in the control group) and the interpretational difficulties we did not use this measure in regression models focusing on predictors clearly linked to inhibitory deficits.

5. The contribution of the interaction term in the regression is unclear, the authors should report the R square change due to the interaction term.

We have added the missing R2 change values to the results section (Tables 6. and 7).

Reviewer 2 Report

The study investigates cognitive functions in ADHD and healthy control children in association with body weight. The ADHD children had no significantly higher BMIs, but performed significantly worse in the cognitive tests as expected. The authors hinted at worse cognitive performance in children with higher BMI but not specifically in ADHD children. 

The study is of interest, however, I have some suggestions. 

  1. instead of "people", I would rephrase using "children/adolescents" or "adults" in the whole manuscript for clearer readability
  2. The authors tested several variables in a relatively small sample, did they correct for multiple comparison? A post hoc power analysis would also be enlightning concerning the main effects, ADHD and BMI on cognitive functioning
  3. It would be enlightning to also have a table with the raw BMI data in addition to the standardized mean values. 
  4. The tables all need legends containing abbreviations and short descriptions of what was done and what was shown because it is really hard to read them and understand them at once
  5. The same for the figures: legends are needed with abbreviations explained and short descriptions what was done and what is shown. 
  6. Conclusions: line 310: why do the results suggest that obesity in ADHD might be rather associated with impulsivity than attention deficit? How can the authors conclude that from their data? 
  7. The authors talk a lot about obesity and ADHD in the conclusions but in their study they could not confirm that the ADHD children had higher BMI than the control children. So they should relate more closely to their actual findings in the discussion and conclusion section, not to what other studies could show previoulsy.  

Author Response

Dear editors and reviewers,

We would like to thank you for all the insightful comments on our manuscript “The association between executive functions and body weight/BMI in children and adolescents with ADHD”. All of these comments were very helpful and guided our efforts to substantially revise and improve our paper. We did our best to make relevant corrections. The detailed responses are provided below each of the reviewers’ remarks.

  1. instead of "people", I would rephrase using "children/adolescents" or "adults" in the whole manuscript for clearer readability

Thank you for that suggestion. We have implemented it throughout the text.

  1. The authors tested several variables in a relatively small sample, did they correct for multiple comparison? A post hoc power analysis would also be enlightning concerning the main effects, ADHD and BMI on cognitive functioning

The key hypotheses, concerning the role of the executive functions in shaping AHDH-bodyweight relationship, were theoretically driven, as already described in the response to the remarks by the first reviewer.

Referring to the question of power-analysis, we conducted some tests prior to the beginning of the project. Our design offered satisfactory power (close to 80%) in detecting moderating effects of f2 = 0.06 (alpha = 0.05). Calculations (conducted in G*power) were based on the assumption that one extra effect is added to a model containing 3 predictors and R2 for the full model equals 0.20. As these computations were based on some arbitrary assumptions, and no relevant literature was available, we decided not to include them into the paper.

  1. It would be enlightning to also have a table with the raw BMI data in addition to the standardized mean values. 

We added basic between-group comparisons and descriptive statistics to the results section (Table 3.).

  1. The tables all need legends containing abbreviations and short descriptions of what was done and what was shown because it is really hard to read them and understand them at once
  2. The same for the figures: legends are needed with abbreviations explained and short descriptions what was done and what is shown. 

Thank you for these suggestions. We have updated all figure and and table descriptions accordingly.

  1. Conclusions: line 310: why do the results suggest that obesity in ADHD might be rather associated with impulsivity than attention deficit? How can the authors conclude that from their data? 

We have added explanation to the text:

Rapid-response impulsivity also referred to as response inhibition or impulsive action is one of the dimensions of impulsivity [55], which characterize ADHD and is investigated using go/no go task like in SART. In our study subjects with higher weight in the ADHD group presented a lower efficiency of the inhibition processes and gave more impulsive and incorrect answers in SART, which may suggest that the problem of excessive weight is associated with impulsivity symptoms. In contrast, no similar relationship was detected for executive score measured by ANT.

  1. The authors talk a lot about obesity and ADHD in the conclusions but in their study they could not confirm that the ADHD children had higher BMI than the control children. So they should relate more closely to their actual findings in the discussion and conclusion section, not to what other studies could show previoulsy. 

Thank you for these suggestions. We have changed the discussion and conclusion part accordingly.

Round 2

Reviewer 1 Report

Thank the authors for addressing my concerns.